# Rapid Size-Dependent Impact of Cu and CuO Nanoparticles on Lentil Seeds and Leaves Using Biospeckle Optical Coherence Tomography

**DOI:** 10.3390/nano15161214

**Published:** 2025-08-08

**Authors:** Lavista Tyagi, Hirofumi Kadono, Uma Maheswari Rajagopalan

**Affiliations:** 1Graduate School of Science and Engineering, Saitama University, 255 Shimo-Okubo, Sakura-Ku, Saitama 338-8570, Japan; tyagi.l.068@ms.saitama-u.ac.jp; 2Innovative Global Program, Shibaura Institute of Technology, 3-7-5 Toyosu, Koto-ku, Tokyo 135-8548, Japan

**Keywords:** copper oxide, metal copper, lentil, nanomaterial, biospeckle, optical coherence tomography

## Abstract

Significant concerns regarding the impact of copper (Cu) and copper oxide (CuO) nanoparticles (NPs) and microparticles (MPs) on plant systems have been brought to light through the growing use of these materials in industry and agriculture. The properties of NPs are critical in determining their uptake by plant cells and the ensuing effects on plant physiology. This emphasizes the need for accurate monitoring techniques to determine the impact caused by NPs on seed development and plant growth. This study uses foliar exposure at 0 and 100 mg/L, as well as seed exposure at 0, 25, and 100 mg/L, to explore the effects of Cu (<10–25 μm; 25 nm) and CuO (<10 µm; <50 nm) NPs and MPs on lentil (*Lens culinaris*). Biospeckle optical coherence tomography (bOCT) was employed to monitor internal physiological activity in real time, non-invasively—capabilities that static imaging methods, such as OCT, are unable to provide. Results showed that exposure to Cu and CuO NPs led to significant reductions in biospeckle contrast, indicating heightened physiological stress, while MPs generally produced minimal or even positive effects. These early changes detected by bOCT within just 6 h of exposure were consistent with traditional morphological and biochemical assessments—such as germination rate, growth, biomass, and catalase activity—that typically require several days to detect. The study demonstrates that bOCT enables the rapid, functional assessment of nanomaterial effects, including those resulting from foliar exposure, thereby offering a powerful tool for early and non-destructive evaluation of plant responses to engineered particles in agricultural contexts.

## 1. Introduction

Among the most widely produced nanomaterials, copper-based nanoparticles (NPs), which include elemental copper (Cu) and copper oxide (CuO), find extensive applications in industries such as environmental remediation, electronics, medicine, and agriculture. By 2034, the market for nano-CuO alone is anticipated to grow from over USD 249.76 million in 2025 to exceed USD 1.3 billion, demonstrating its increasing industrial significance [1]. These NPs are particularly promising as nano-fertilizers, since Cu is essential for key plant functions, including photosynthesis, respiration, and enzyme activity. However, the growing popularity of Cu-based NPs has raised concerns regarding their environmental persistence and bioaccumulation potential, particularly in relation to plant interactions [2].

Copper-based nanoparticles (Cu-based NPs) possess unique physicochemical properties that influence their interactions with plants, largely depending on particle size and concentration. According to Yang et al. [3], low doses of these NPs can promote callus formation and regeneration in crops such as rice (*Oryza sativa*), though smaller particles often exhibit heightened reactivity and toxicity. In contrast, higher concentrations (500–1000 mg/L) have been shown to inhibit root elongation, reduce biomass, and impair photosynthetic efficiency in species like maize, wheat, and rice [4,5]. These adverse effects correlate with the accumulation of Cu-based NPs in chloroplasts, where they compromise thylakoid membrane integrity and decrease photosystem II photochemical efficiency by up to 46%. Markers of oxidative stress, such as elevated malondialdehyde levels and altered antioxidant enzyme activity, further highlight the phytotoxic potential of these NPs. Additionally, Cu-based NPs disrupt the uptake of essential nutrients, leading to diminished absorption of zinc, calcium, and iron [3,6,7].

Traditional methods for evaluating NP-induced plant responses typically involve destructive analyses performed after visible growth has occurred. These include measuring seed germination rates, biomass accumulation, and seedling length and conducting biochemical assays for antioxidant enzymes and reactive oxygen species (ROS). Uptake and localization of NPs are often visualized using transmission or scanning electron microscopy. However, these techniques are invasive, do not permit real-time monitoring and may overlook subtle physiological changes occurring during early exposure. Consequently, conventional approaches fail to capture NP effects during the critical pre-germination phase, leaving a gap in understanding early plant–NP interactions.

Optical coherence tomography (OCT), a non-invasive imaging modality, offers high-resolution cross-sectional and three-dimensional views of internal tissue architecture. Unlike traditional microscopy, OCT enables the real-time visualization of sub-surface structural features without harming the sample, making it increasingly valuable in plant science applications [8].

We employed Biospeckle Optical Coherence Tomography (bOCT), a dynamic imaging technique that integrates biospeckle analysis with conventional OCT to monitor internal tissue activity. By capturing dynamic responses within plant tissues, bOCT enables the real-time, non-invasive detection of subtle physiological changes, even prior to visible germination [9,10]. Our research group has successfully applied this technique to visualize the effects of alumina nanoparticles on lentil (*Lens culinaris*) seeds, as well as to monitor the impacts of heavy metal exposure, such as zinc, within as little as six hours. Due to its high sensitivity and temporal resolution, bOCT offers a powerful tool for detecting and tracking nanoparticle-induced changes in seeds, effectively addressing a key limitation in current phytotoxicity assessment methods [11,12].

The present study investigates the effects of Cu and CuO NPs on lentil, a nutritionally valuable crop. Owing to its high protein and flavonoid content, lentil plays a crucial role in dietary supplementation and nutritional security across many Asian countries. Its rapid germination, distinctive seed coat structure, and broad environmental adaptability also make it an ideal model for plant-based nanotoxicological studies [13,14]. In addition to employing bOCT for real-time imaging, we assessed conventional phytotoxicity endpoints, including seedling growth, biomass accumulation, and physiological responses, specifically catalase (CAT) activity, after seven days of exposure.

Recognizing the significance of above-ground exposure pathways, we also conducted foliar applications of Cu and CuO NPs on lentil leaves and monitored internal physiological activity using bOCT for up to 48 h. Leaf exposure is particularly relevant, as foliage represents a primary interface for NPs contact in agricultural environments. This is increasingly important given the rising use of foliar nano-fertilizers, nano-pesticides, nano-sensors, and nanocarriers in the agriculture sector [15]. Utilizing bOCT, we were able to non-invasively detect early and subtle changes in internal leaf activity following NP exposure, providing a comprehensive view of plant–NPs interactions under foliar treatment conditions.

## 2. Resources and Techniques

### 2.1. Cu-Based NPs and Sample Preparation

We used copper oxide (CuO) and copper (Cu) metal particles obtained as dry powders from Merck Sigma-Aldrich (Tokyo, Japan). The CuO particles included <50 nm primary particle size (TEM, ≥99.5%) and <10 µm particles (≥99.9% trace metal basis), while the Cu metal particles were 25 nm and 10–25 µm in size. Detailed physicochemical properties of each particle type, as specified in the certificates of analysis provided by the supplier, along with scanning electron microscopy (SEM) images, are presented in Appendix A.

A 0.05% Tween-80 solution was used as a non-ionic, biocompatible surfactant to address the hydrophobic character of the particles and encourage uniform dispersion. No further purification was performed prior to dispersion. After 30 min of initial stirring with a magnetic stirrer, the dispersion method involved sonication for 20 min at 28 °C in low-noise mode at 100% amplitude. To make sure there were no particle clusters in the solutions, this procedure was repeated until a uniform dispersion was achieved. According to the supplier’s certificate of analysis, the levels of trace impurities are minimal and within the specifications for analytical-grade materials. While commercial NPs are widely used in plant nanotoxicology studies due to their high purity and reproducibility, we acknowledge that trace impurities cannot be entirely excluded without additional purification or analytical verification.

To facilitate uniform dispersion and counteract the hydrophobicity of the particles, a 0.05% Tween-80 solution—a non-ionic, biocompatible surfactant—was used. No additional purification was performed prior to dispersion. Initially, the solutions containing dry powders were stirred for 30 min using a magnetic stirrer. This was followed by sonication at 28 °C in low-noise mode at 100% amplitude for 20 min. The procedure was repeated until a visually uniform dispersion was achieved, minimizing the likelihood of particle agglomeration. According to the supplier’s analysis, trace impurity levels were minimal and within analytical-grade specifications. While commercial NPs are widely used in plant nanotoxicology studies due to their high reproducibility and purity, we acknowledge that trace impurities may still be present in small amounts unless further purification or verification is conducted.

Organic lentil (*Lens culinaris*) seeds were supplied by Greenfield Project Co. Ltd. (Tokyo, Japan). Seeds were stored in a dry, refrigerated environment until use. Prior to treatment, seeds (~30 mg each) were carefully selected for uniformity and surface sterilized by immersion in a 2.5% hydrogen peroxide (H_2_O_2_) solution for 10 min, followed by rinsing them with three rinses with purified water. This sterilization protocol is widely used and has been shown not to interfere with germination, as H_2_O_2_ rapidly decomposes into water and oxygen, and residual effects are minimized through thorough rinsing [16].

Following sterilization, seeds were grouped into sets of six. Treatments included exposure to CuO and Cu (both NPs and microparticles (MPs)) at concentrations of 25 and 100 mg/L, while control groups received distilled water. Seed samples were incubated in petri dishes within a growth chamber (MLR-351H, SANYO Electric Co., Ltd., Osaka, Japan) under controlled conditions: a 12 h light/dark cycle, 50–65% relative humidity, day/night temperatures of 25 °C/20 °C, and light intensities of 260–370 μmol m^−2^ s^−1^ (day) and 0 μmol m^−2^ s^−1^ (night). Biospeckle Optical Coherence Tomography (bOCT) images were captured at 0, 6, 24, and 48 h after treatment to monitor physiological changes in real time.

For foliar exposure experiments, the same procedure was followed using 15-day-old lentil plants. A 1 mL dispersion of CuO or Cu particles (NPs and MPs) at 100 mg/L was uniformly sprayed onto the leaf surface of each plant. Internal leaf activity was then monitored using bOCT over the same 48 h observation period.

### 2.2. Biospeckle Optical Coherence Tomography (bOCT) Overview

#### 2.2.1. bOCT Experimental System

Figure 1 presents a schematic of the fiber-based, custom-built swept-source Optical Coherence Tomography (OCT) system developed on an optical bench. This system was employed to monitor internal biological activity in lentil seeds within just a few hours of exposure to various NPs treatments. A brief overview of the OCT system is provided here; detailed specifications and technical validation are available in our previous study on the effects of ZnO NPs and MPs on lentils [17]

The OCT system utilizes a swept-source laser with a central wavelength of 1.3 µm, a spectral bandwidth of 125 nm, an average output power of 23.4 mW, and a sweep frequency of 20 kHz. This configuration yields an axial resolution of 6 µm and a lateral resolution of 39 µm. During imaging, the incident power on the sample was maintained at approximately 100 µW, with an exposure duration of 8 s and a corresponding radiant dose of 1.39 mJ/cm^2^—well below thresholds associated with biological tissue damage. Data acquisition and image processing were conducted using custom-developed routines in LabVIEW (LabVIEW ver.2012 National Instruments, Austin, TX, USA) and MATLAB (MathWorks version R2016b, Natick, MA, USA).

The safety of this imaging protocol is supported by prior studies, such as that by Hasan et al., which reported no adverse effects on maize seeds even at irradiation intensities up to 4 mW/cm^2^ for durations up to 105 s [18].

In conventional OCT, artifacts such as segmentation errors, shadowing, mirror artifacts, and motion-induced distortions are well documented and can impact image interpretation. In the context of bOCT, the speckle pattern is intentionally analyzed as a signal reflecting internal biological activity, rather than being treated solely as noise. However, other sources of artifacts, such as sample motion, optical misalignment, or system noise, may still influence the speckle signal and should be considered when interpreting results. To minimize sample motion and associated artifacts, all samples were mounted on a vibration-isolated optical table, and care was taken to avoid any external vibrations during image acquisition. Additionally, to reduce variability between seeds and ensure robust quantitative analysis, biospeckle contrast values were calculated for multiple regions of interest (ROIs) within each seed, and data were normalized to pre-imbibition values for each sample. This two-step procedure helped control for both technical and biological variability, allowing for more accurate assessment of physiologically meaningful speckle fluctuations.

#### 2.2.2. Biospeckle Fundamentals and Contrast Analysis

When a plant is illuminated with coherent light, its optically rough surface features scatter the light and produce a granular interference pattern known as biospeckle. This pattern arises from the random interference of scattered light waves [19]. For stationary objects, the speckle pattern remains temporally stable. In contrast, dynamic biological samples—such as plant seeds—exhibit time-dependent intensity fluctuations in the speckle pattern due to continuous internal motion. These include processes like cytoplasmic streaming, cellular proliferation, water transport, and tissue growth.

The term biospeckle specifically refers to these temporal intensity variations, which are indicative of internal biological activity. In bOCT structural images, such variations manifest as intensity changes at each pixel over time, enabling the visualization of dynamic processes within plant tissues.

To quantify this activity, biospeckle contrast (*γ*) is calculated. It is defined as the ratio of the standard deviation (*σ*) to the mean (*μ*) of the pixel intensity over a temporal sequence during scanning. This contrast value provides a pixel-wise measure of the degree of fluctuation, where higher values correspond to greater internal biological activity. The following equation is used to calculate this contrast,(1)$$\gamma \;\left(x,z\right)=\frac{1}{<I_{OCT}\left(x,z\right)>}{\left[\frac{1}{N}\;\sum_{j=1}^{N}{\left\{I_{OCT}\left(x,z,\;t_{j}\right)-<I_{OCT}\left(x,z\right)>\right\}}^{2}\right]}^{\frac{1}{2}}$$
where$${<I}_{OCT}\left(x,z\right)>=\;\frac{1}{N}\;\sum_{j=1}^{N}I_{OCT}\left(x,z,\;t_{j}\right)$$

The lateral coordinate is represented by *x* in the preceding equation, the depth coordinate by *z*, the scan number by *j*, the time associated with the *jth* scan by *t_j_*, and the total number of scans by *N*. Greater temporal intensity fluctuations are reflected in a larger biospeckle contrast value, which indicates more internal activity in the seed under study. Conversely, smaller temporal fluctuations are associated with a lower biospeckle contrast, suggesting less internal activity. As a result, biospeckle contrast is a useful characteristic for assessing how seeds react to their external environment.

### 2.3. Conventional Measures

#### 2.3.1. Growth Indicators

The radicle of a lentil seed was considered germinated when it extended approximately 2 mm, usually within a day. Using ImageJ software (v1.54d, NIH, Bethesda, MD, USA), root and shoot lengths of seedlings grown from three replicates of ten seeds per treatment, including Cu and CuO NPs and MPs at the same bOCT observation concentrations, were measured. Germination percentage was calculated to assess seed performance using the given equation.(2)$$Germination\;Percentage\;\left(\%\right)=\frac{Germinated\;seeds}{total\;seeds}\times 100$$

Seedlings were carefully cleaned and separated into roots and shoots before fresh weight measurements were taken using an analytical balance (AUX 320, UniBloc, Shimadzu Corporation, Kyoto, Japan). For dry weight determination, samples were oven-dried initially at 105 °C for two hours, followed by drying at 80 °C until a constant weight was achieved, typically over 72 h (SOFW-450S, AS ONE, Osaka City, Japan). By correlating these biomass measurements with early-stage bOCT observations made prior to germination, we gained valuable insights into plant water content and growth responses under varying treatments of Cu-based NPs.

#### 2.3.2. Biochemical Indicators

After seven days of germination, antioxidative enzyme activity in lentil seedlings exposed to Cu-based NPs was assessed to compare the bOCT results with conventional biochemical assays. Seedling samples were rinsed and homogenized in ice-cold phosphate buffer (0.01 mol/L, pH 7.4) using a pre-chilled mortar and pestle. The homogenates were centrifuged at 3000 rpm for 15 min at 4 °C, and the resulting supernatants were collected for enzyme analysis.

Absorbance measurements were performed using a spectrophotometer (V-730, JASCO Corporation, Tokyo, Japan) with a spectral bandwidth of 1 nm to ensure high resolution over a wavelength range of 190–1100 nm. CAT activity was determined following Aebi’s method (1984) by monitoring the decrease in absorbance at 240 nm over a 3 min interval, recorded every minute [20]. This decrease corresponds to the enzymatic breakdown of H_2_O_2_ by CAT.

### 2.4. Statistical Analysis

Initial OCT image data acquisition was performed using custom software developed in LabVIEW (version 2012, National Instruments, Austin, TX, USA). Subsequent biospeckle OCT data analysis was conducted in MATLAB (R2016b). Graphs and histograms were generated using Origin 9.5, while statistical analyses were carried out in Microsoft Excel (Microsoft 365, version 2412, USA). Results are presented as mean ± standard deviation, and statistical significance between groups was determined using Student’s *t*-test, with a threshold of *p* < 0.05.

## 3. Results

### 3.1. bOCT Results

#### 3.1.1. Impact of CuO Particles

The impact of CuO particles on lentil seed internal activity was evaluated through both qualitative and quantitative bOCT analyses. A comprehensive comparison between bOCT and conventional OCT imaging, along with qualitative assessments, has been detailed in our previous study on the effects of ZnO NPs on lentil seeds [17].

Qualitatively, Figure 2a presents bOCT contrast images depicting internal seed activity following exposure to CuO particles of different sizes at a concentration of 100 mg/L. In these color-coded images, red regions indicate high biospeckle contrast values, reflecting elevated internal biological activity, while blue regions correspond to low activity. The matrix layout of Figure 2a organizes images by exposure time (0, 6, 24, and 48 h) and particle size (CuO MPs: <10 µm; CuO NPs: <50 nm), alongside a control group.

A distinct trend is observed where seeds treated with nano-sized CuO particles exhibit a greater prevalence of blue regions compared to those exposed to micro-sized particles, particularly as exposure duration increases. This visual evidence suggests that nano-sized CuO particles more effectively suppress internal seed activity relative to their micro-sized counterparts.

Quantitative support for these observations is provided by the averaged normalized contrast (ANC) results presented in Figure 2b,c. ANC is defined as the mean of normalized contrast values, calculated by first determining the biospeckle contrast for each ROI using Equation (1), then normalizing each value to the pre-imbibition baseline (0 h) for individual seeds, and finally averaging across all seeds within each treatment group.

At the lower concentration of 25 mg/L (Figure 2b), micro-sized CuO particles induced a significant increase in ANC after 6 h, suggesting a transient stimulation of internal seed activity. In contrast, nano-sized CuO particles elicited a pronounced decrease in ANC, indicating an inhibitory effect on seed physiological processes. At the higher concentration of 100 mg/L (Figure 2c), this inhibitory effect of nano-sized CuO particles was further amplified, with ANC values declining markedly after 48 h. Meanwhile, micro-sized CuO particles exhibited no significant impact on ANC relative to the control at this concentration.

#### 3.1.2. Impact of Cu Metal Particles

For Cu metal particles, both qualitative and quantitative bOCT analyses reveal distinct patterns influenced by particle size and concentration. Qualitatively, the bOCT images (Figure 3a) indicate that seeds treated with nano-sized Cu particles consistently exhibit a predominance of blue coloration, particularly at the higher concentration of 100 mg/L and longer exposure duration (48 h), reflecting a substantial reduction in internal biological activity. In contrast, seeds exposed to micro-sized Cu particles display more extensive red regions, suggesting that internal activity is maintained or even enhanced under certain conditions.

These visual observations are supported by quantitative ANC analyses. Exposure to nano-sized Cu particles at both 25 mg/L and 100 mg/L concentrations led to a significant decrease in ANC values, confirming a strong inhibitory effect on seed internal activity (Figure 3b,c). Conversely, micro-sized Cu particles elicited a significant increase in ANC after 24 h at both concentrations, indicating a potential stimulatory influence.

### 3.2. Comparison with Conventional Measurements

#### 3.2.1. Biochemical Indicators

After seven days of exposure, CAT enzyme activity was measured in lentil seedlings treated with both CuO (Figure 4a) and Cu (Figure 4b) NPs and MPs. The vertical axis represents CAT activity expressed in units per gram per minute (U/g·min), while the horizontal axis indicates particle concentration in mg/L.

In Figure 4a, at 25 mg/L, seeds exposed to micro-sized CuO particles showed a decrease in CAT activity relative to the control. Although the CAT levels did not exceed those observed in the NP-treated group, the increase at 100 mg/L was only slight compared to the control. In contrast, exposure to CuO NPs induced a significant and pronounced increase in CAT activity at both 25 mg/L and 100 mg/L compared to untreated seeds.

Similarly, in Figure 4b, exposure to micro-sized Cu particles resulted in a significantly lower CAT content at 25 mg/L than in the control. When the concentration was raised to 100 mg/L, CAT content increased compared to 25 mg/L but remained below that of the NPs’ treatment at the same concentration. Seeds exposed to Cu NPs showed a substantial and significant elevation in CAT content at both concentrations relative to the control.

Overall, for both types of micro particles, CAT content decreased at the lower concentration and partially recovered at the higher concentration but consistently remained lower than in the NP-treated seeds, where the increase was both higher and statistically significant compared to the control at higher concentration.

#### 3.2.2. Growth Indicators

After seven days of exposure to CuO and Cu NPs and MPs at concentrations of 25 mg/L and 100 mg/L, the root and shoot lengths of lentil seedlings were measured (Figure 5a–f). CuO NPs significantly reduced both root and shoot lengths at 100 mg/L compared to the control, with a more pronounced reduction observed in root length (Figure 5a).

In contrast, CuO MPs at the same concentration did not significantly affect root or shoot lengths relative to the control. At 25 mg/L, CuO NPs induced a moderate decrease in both root and shoot lengths, whereas CuO MPs led to shoot lengths exceeding those of the control and root lengths comparable to the control group (Figure 5b). Representative images of lentil seedlings exposed to CuO NPs and MPs are shown in Figure 5c.

Exposure to Cu NPs at 100 mg/L resulted in a notable reduction in both root and shoot lengths compared to the control, with root length being more severely affected (Figure 5d). In contrast, Cu MPs at the same concentration had no significant impact on either root or shoot growth. At the lower concentration of 25 mg/L, both Cu NPs and MPs produced root and shoot lengths comparable to those of the control group (Figure 5e). Representative images of lentil seedlings treated with Cu NPs and MPs are presented in Figure 5f.

The fresh and dry weights of the roots and shoots of lentil seedlings subjected to 25 mg/L and 100 mg/L of CuO MPs and NPs were also measured (Figure 6a–d). CuO NPs at 100 mg/L decreased the fresh and dry weights of roots and shoots, with the effect on roots being more prominent (Figure 6a,b). Compared to the control, CuO MPs at this concentration resulted in significantly less weight loss. CuO MPs raised root weights and did not affect shoot weights at 25 mg/L, whereas CuO NPs slightly decreased fresh and dry weights (Figure 6c,d).

For Cu treatments (Figure 7a–d), Cu NPs at 100 mg/L led to substantial reductions in both fresh and dry weights of roots and shoots, with the effect being especially pronounced in roots. Cu MPs at this concentration did not significantly alter weights relative to the control group. At 25 mg/L, Cu NPs caused moderate decreases in these weights, whereas Cu MPs resulted in weights that were comparable to, or slightly greater than, those of the control.

Overall, roots were more impacted than shoots when exposed to CuO and Cu NPs at 100 mg/L, which resulted in the largest decreases in fresh and dry weights as well as root and shoot lengths. MPs had little or no negative effect, and in some cases, at 25 mg/L, root weights were higher than those of the control.

### 3.3. bOCT Leaf Exposure Results

Leaf exposure to Cu and CuO NPs and MPs at 100 mg/L revealed notable differences compared to seed exposure (Figure 8a,b). As shown in Figure 8a, CuO NPs had a rapid and significant inhibitory effect on internal leaf activity, with ANC values decreasing markedly just 6 h after exposure. In contrast, CuO MPs did not produce any significant change in ANC, even at the same concentration.

Interestingly, Figure 8b shows that both Cu NPs and MPs elicited a stimulatory effect on lentil leaves. ANC values increased significantly for both particle sizes within 6 h of exposure, indicating enhanced internal physiological activity. These contrasting responses underscore the importance of both particle composition and exposure route (seed vs. leaf) in determining NP–plant interactions.

## 4. Discussion

The objective of this study was to compare the effects of CuO and metallic Cu NPs and MPs on the initial and early growth of lentil seeds. We used bOCT, a non-invasive imaging method, to rapidly and safely observe internal activity within seeds and seedlings during growth. These observations were correlated with traditional growth metrics, including germination rate, root shoot elongation, biomass production, and antioxidative enzyme responses. While standard OCT generates structural images, bOCT uniquely captures real-time dynamic changes in seeds during germination in vivo, enabling detailed analysis of NPs effects on developmental processes within much shorter timeframes than traditional approaches.

A summary of the effects of Cu- based NPs and MPs at 25 and 100 mg/L on lentil seed and leaves by bOCT measurements (24 h) and the conventional physiological and biochemical parameters (7 days) is presented in Table 1. The table provides an overview of the observed changes relative to the control across all measured endpoints.

The bOCT technique enabled the rapid and sensitive detection of internal physiological changes in lentil seeds following exposure to Cu and CuO particles. Significant changes in ANC values were observed as early as 6 h post-exposure, with the most pronounced decreases occurring in seeds treated with NPs, particularly at higher concentrations (100 mg/L) shown in Figure 3a–c. This early decline in ANC suggests that NPs rapidly disrupt internal seed activity, likely by interfering with metabolic processes essential for germination, as the pore size of lentil seed is approximately 4–8 µm, which allows NPs to enter inside the seeds [21].

In contrast, MP treatments induced either minimal changes or slight increases in ANC at lower concentrations, indicating a milder or even stimulatory effect on seed internal activity [4,22]. These findings are consistent with previous studies of Cu-based NPs, demonstrating the sensitivity of bOCT in detecting early physiological responses to NPs exposure, thus highlighting its potential as a non-invasive method for the real-time monitoring of seed health [4].

The conventional morphological measures further substantiated the bOCT findings. Exposure to Cu and CuO NPs, especially at 100 mg/L, resulted in significant reductions in root and shoot lengths, as well as fresh and dry biomass, with root growth being particularly affected (Figure 5, Figure 6 and Figure 7). These negative effects are likely attributable to the enhanced bioavailability and reactivity of NPs, which facilitate their penetration into seed tissues and subsequent disruption of cellular functions [23,24,25].

In contrast, MPs generally exhibited negligible or even positive effects on seedling growth at lower concentrations, possibly acting as a micronutrient source rather than a stressor. This size-dependent toxicity aligns with the broader literature on metal oxide particles in plants, where smaller particles are more likely to penetrate biological barriers and induce physiological stress [26].

Our previous investigation into the effects of ZnO NPs and MPs on lentil seeds similarly demonstrated the negative impact of metal oxides at smaller particle sizes [17], further supporting the findings of this study. However, contrasting results were observed in our recent study on TiO_2_ NPs, where both nano- and micro-sized particles exhibited beneficial effects on lentil seed germination and growth, even at concentrations up to 200 mg/L. This highlights the complex and material-specific nature of NP–plant interactions. These findings suggest that the effects of NPs on plants are not governed solely by particle size or concentration, but also by the intrinsic physicochemical properties of the material at the nanoscale. Such variability underscores the need for case-by-case evaluation of different nanomaterials, as the same plant species may respond differently depending on the NP composition [Under review].

The biochemical analysis of CAT activity provded further insight into the oxidative stress responses elicited by particle exposure (Figure 4a,b). Seeds exposed to Cu and CuO NPs exhibited a marked and statistically significant increase in CAT activity at both 25 mg/L and 100 mg/L compared to the control and MP treatments. This upregulation of CAT activity is indicative of an enhanced antioxidant defense response, likely triggered by an increased production of ROS in NP-treated seeds.

Conversely, seeds treated with MPs showed decreased CAT activity at 25 mg/L and only partial recovery at 100 mg/L, with levels remaining lower than those observed in NP treatments. This suggests that MPs induce less oxidative stress, consistent with their limited cellular penetration and lower reactivity. These results mirror findings from previous research, where NPs exposure led to increased antioxidant enzyme activity as a protective mechanism against oxidative damage [27,28,29].

The observed negative effects of Cu and CuO NPs on lentil seed development can be attributed to well-established biological mechanisms by which these NPs induce oxidative stress in plants. Due to their small size, Cu and CuO NPs are capable of penetrating seed pores and entering plant tissues, where they can release Cu^2+^ ions and participate in redox cycling reactions. These processes generate ROS, including superoxide anions (O_2_^−^) and H_2_O_2_, which disrupt cellular redox balance.

The overaccumulation of ROS leads to oxidative damage of cellular components such as lipids, proteins, and nucleic acids, impairing essential physiological functions. This oxidative stress manifests as observable phenotypic changes, including reduced root and shoot growth, compromised membrane integrity, and diminished photosynthetic efficiency, as documented in earlier studies [5,30,31].

Additionally, Cu NPs have been reported to disrupt the photosynthetic apparatus, particularly photosystem II, leading to altered chlorophyll fluorescence and increased non-photochemical quenching, manifested as energy dissipation in the form of heat. These changes are characteristic indicators of oxidative stress. In response to elevated levels of ROSs, plants activate their antioxidant defense systems, including enzymes such as CAT, to mitigate cellular damage.

However, when ROS generation surpasses the capacity of these antioxidant defenses, oxidative damage accumulates, resulting in impaired cellular function and inhibited growth. This mechanistic understanding aligns closely with our findings from both bOCT and conventional measurements, providing a comprehensive view of the phytotoxic effects induced by CuO NPs [32,33,34].

The results of foliar exposure presented in Figure 8a,b demonstrate that the effects of NPs on lentil leaves are influenced not only by particle size and concentration but also by the specific tissue exposed. Cu metal NPs appeared to have beneficial effects on lentil leaves, potentially functioning as micronutrients and enhancing photosynthetic activity, consistent with findings reported by Rameen et al. in *Brassica napus* L. [35]. In contrast, CuO NPs exhibited pronounced phytotoxicity, likely due to their capacity to block stomatal pores, which range from ~12 μm in length and ~2 μm in width for lentil leaves, thereby disrupting gas exchange processes critical for leaf function [36,37,38]. Moreover, NPs can translocate from leaves to roots, potentially causing systemic cellular damage, as observed by Xiong et al. [32].

The contrasting effects observed between Cu metal NPs and CuO NPs can be attributed to their distinct chemical properties and interactions with plant tissues. Cu metal NPs, at moderate concentrations, serve as a source of essential copper ions required for various physiological functions, including photosynthesis and the activation of antioxidant enzymes. This micronutrient role can stimulate internal metabolic activity and enhance plant health, as supported by recent studies demonstrating increased antioxidant capacity and improved stress tolerance in plants treated with Cu NPs [39,40].

In contrast, CuO NPs tend to induce greater oxidative stress and cellular toxicity, largely due to their potent ability to generate ROS at the NP surface and their efficient uptake into plant cells, where they disrupt vital processes such as mitochondrial function. Although CuO NPs release Cu ions in a sustained manner, it is primarily their NP-specific properties—such as surface reactivity, ROS generation, and physical aggregation leading to stomatal blockage—that account for their elevated toxicity, rather than simply a more rapid release of Cu ions [41,42]. Therefore, even at identical concentrations, the chemical form of Cu-based NPs critically determines whether the impact on lentil leaves is beneficial or harmful.

### Influence of Particle Morphology on Plant Responses

Our results indicate that particle morphology, alongside size and composition, plays a critical role in determining the biological impact of Cu-based materials on plant tissues (see Appendix A). The Cu MPs sized 10–25 μm, characterized by a spheroidal shape, consistently exhibited positive effects on both seeds and leaves across all tested concentrations. This was reflected in increased bOCT contrast, seedling length, and biomass. Such larger, smoother, and more uniform spheroidal particles are less likely to penetrate plant tissues or cause cellular disruption, potentially facilitating beneficial surface interactions or serving as a slow-release source of copper [43,44].

Similarly, CuO MPs (<10 μm) with an irregular, rectangular morphology demonstrated significant positive effects on seed and seedling growth at the lower concentration of 25 mg/L, as evidenced by enhanced bOCT contrast, seedling length, and biomass. Their larger size and irregular shape may promote advantageous surface interactions or gradual nutrient release without inducing cellular stress, at least up to moderate concentrations.

In contrast, smaller NPs, including CuO NPs (<50 nm, sub-globose [45]) and Cu NPs (~25 nm, spherical), consistently exerted negative effects on seed germination and seedling development. This is likely due to their high surface area and morphology, which facilitate greater reactivity, cellular uptake, and induction of stress responses [46,47]. Notably, Cu NPs (25 nm) exhibited a positive effect on leaf activity at 100 mg/L, potentially owing to their spherical shape that enhances foliar adhesion and promotes physiological processes at the leaf surface. Conversely, CuO MPs at 100 mg/L had no significant effects on either leaves or seeds, possibly because their larger, irregular form limits excessive uptake or toxicity at this concentration.

These findings highlight the importance of morphology, specifically particle shape and surface characteristics, in modulating interactions between Cu particles and plant tissues. These factors influence whether the outcomes are beneficial or adverse, depending on the exposure route and concentration.

The results of this study underscore the critical influence of particle size, morphology, and concentration on the phytotoxicity of metal-based particles. The rapid and pronounced effects of NPs on both internal seed activity and traditional growth parameters highlight their potential risks in agricultural and environmental settings. In contrast, MPs appear to be less disruptive and may even provide beneficial effects at suitable concentrations. The observed elevation of CAT activity in NP-treated seeds further corroborates oxidative stress as a central mechanism driving NP-induced toxicity.

## 5. Conclusions

This study demonstrates bOCT as a powerful analytical technique for monitoring the effects of nano- and micro-sized Cu and CuO particles on the internal activity of seeds as well as leaves of lentil. Within six hours of exposure, the use of bOCT allowed for the quick and non-invasive observation of early seed responses, demonstrating significant changes in internal activity. The findings revealed that Cu and CuO NPs elicit stronger physiological stress in lentil seeds compared to MPs, attributable to their nanoscale dimensions, which facilitate penetration through seed coat microstructures and subsequent interaction with cellular components. This interaction is associated with the generation of ROS and an upregulation of antioxidant defense mechanisms, as evidenced by increased CAT activity in NP-treated seeds.

The size-dependent phytotoxicity of Cu and CuO particles was reflected across multiple parameters, including reductions in root and shoot elongation, biomass accumulation, and changes in enzymatic antioxidant activity. In contrast, MPs generally exerted negligible or even stimulatory effects, likely serving as a gradual source of essential micronutrients without inducing substantial oxidative stress. Additionally, the leaves’ response to Cu and CuO particles was quite different than the seed exposure, as the leaves exposed to CuO NPs had more pronounced negative effects, while metal Cu NPs demonstrated a positive impact, suggesting that not just size and dose but exposed tissue also plays a significant role in NP–plant interactions.

The bOCT technique enabled the detection of these size-dependent effects at an early stage, preceding observable changes in the conventional morphological and biochemical indices. Following seven days of exposure, significant effects of both NPs and MPs were further confirmed by conventional physiological and biochemical analyses, highlighting the bOCT approach’s sensitivity, reliability, and scientific significance. Importantly, bOCT facilitated the dynamic, in vivo monitoring of internal activity of the seed during germination, providing functional information beyond the structural imaging capabilities of the conventional OCT.

These results underscore the critical influence of particle size and concentration on the phytotoxicity of metal-based materials in plant systems. The substantial negative impacts of NPs draw attention to possible risks resulting from their use in agriculture, whereas MPs may offer safer or even beneficial outcomes under certain conditions. Further research is required to elucidate the long-term consequences of these interactions and in order to optimize the bOCT methodology for wider use in ecological and agronomic assessment.

## Figures and Tables

**Figure 1 nanomaterials-15-01214-f001:**
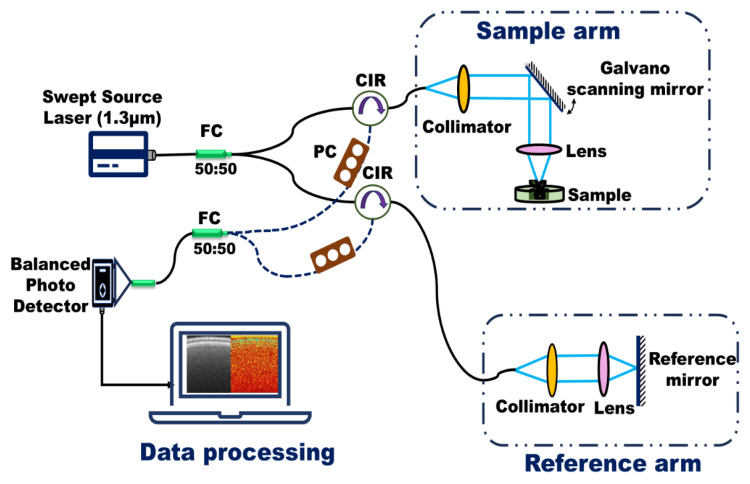
Swept-Source Optical Coherence Tomography (SS-OCT) system used in experiments (FC: Fiber Coupler, CIR: Circulator, PC: Polarization Controller), the same as utilized in our earlier study [17].

**Figure 2 nanomaterials-15-01214-f002:**
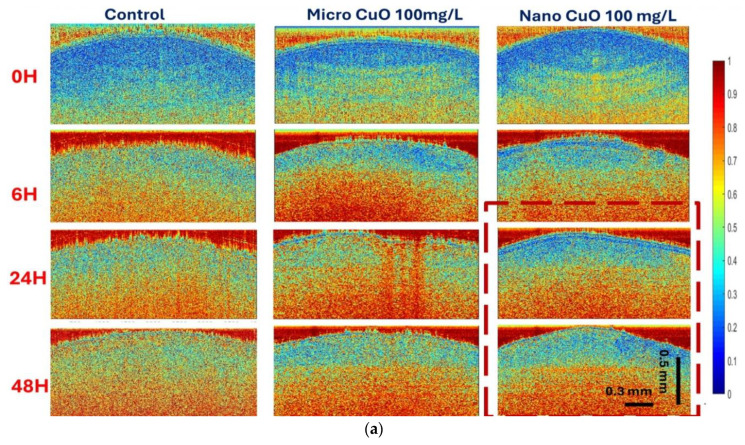
(**a**) bOCT contrast images of lentil seeds exposed to CuO particles at 100 mg/L, shown at 0, 6, 24, and 48 h (vertical axis: depth; horizontal axis: lateral position; color scale: blue = low, red = high speckle contrast), Red dotted lines highlight the comparison of high blue color densities in seeds exposed to small-sized particles (**b**,**c**) Averaged normalized biospeckle contrast (ANC) of seeds exposed to CuO at (**b**) 25 mg/L and (**c**) 100 mg/L (* denotes the statistical significance of data * *p* < 0.05).

**Figure 3 nanomaterials-15-01214-f003:**
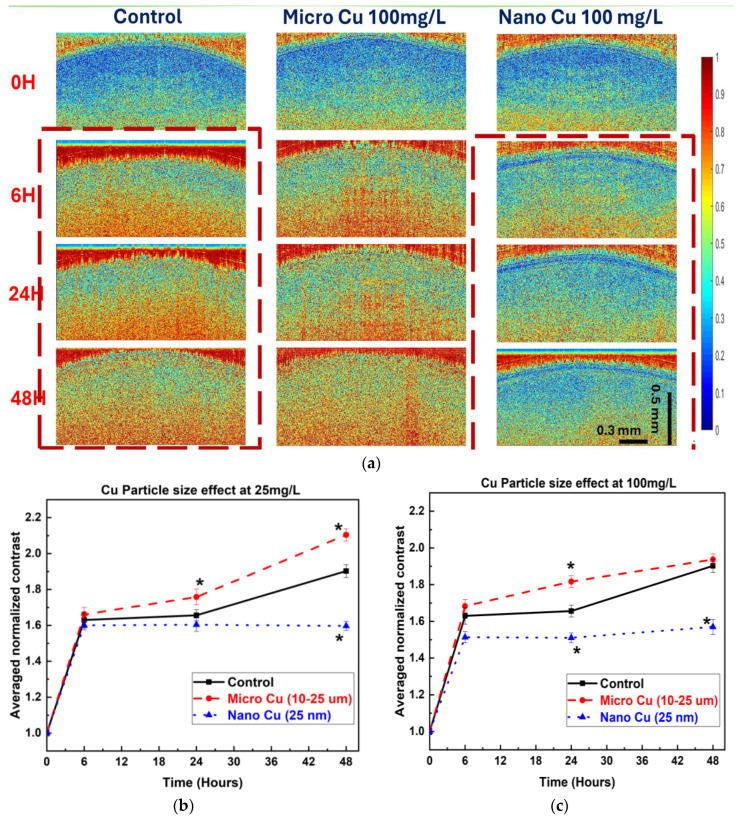
(**a**) bOCT contrast images of lentil seeds exposed to Cu metal particles at 100 mg/L, shown at 0, 6, 24, and 48 h (vertical axis: depth; horizontal axis: lateral position; color scale: blue = low, red = high speckle contrast). Red dotted lines highlight the comparison of high blue and red color densities in seeds exposed to control and nano Cu particles (**b**,**c**) Averaged normalized biospeckle contrast (ANC) of seeds exposed to Cu metal at (**b**) 25 mg/L and (**c**) 100 mg/L (* denotes the statistical significance of data * *p* < 0.05).

**Figure 4 nanomaterials-15-01214-f004:**
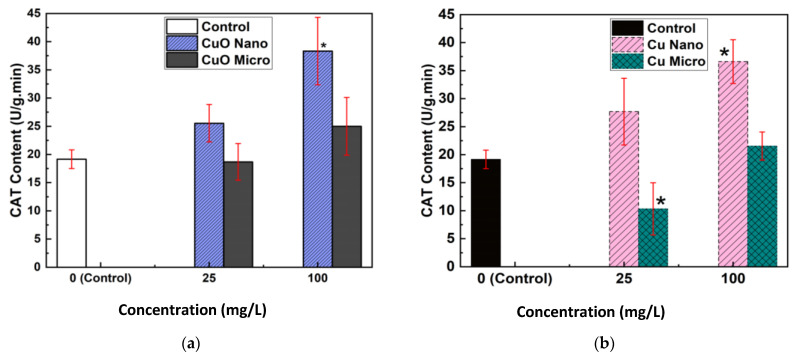
After 7 days, the amount of catalase (CAT) in lentil seedlings exposed to (**a**) CuO and (**b**) Cu particles at 25 and 100 mg/L. * *p* < 0.05 vs. control (*t*-test, statistically significant).

**Figure 5 nanomaterials-15-01214-f005:**
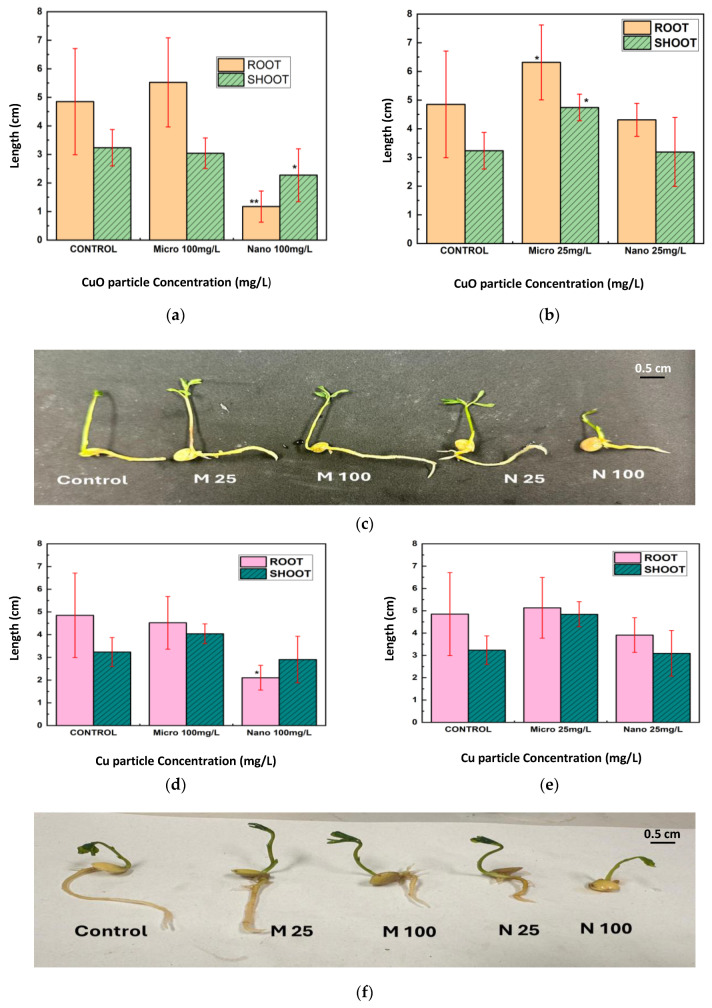
Root and shoot length of lentil seedlings exposed to (**a**,**b**) CuO and (**d**,**e**) Cu particles at 100 and 25 mg/L, (**c**,**f**) photograph of lentil seedlings after 7 days (M25 Micro 25 mg/L, N 25 = Nano 25 mg/L, M100 = Micro 100 mg/L, N 100 = Nano 100 mg/L). Statistical significance was determined by *t*-test; * *p* < 0.05, ** *p* < 0.01 vs. control.

**Figure 6 nanomaterials-15-01214-f006:**
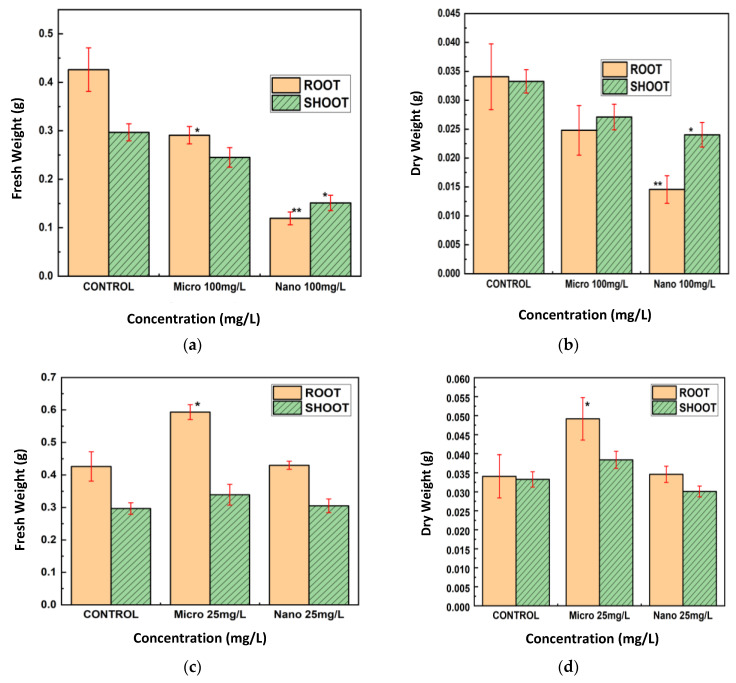
Root and shoot fresh and dry weight, respectively, of lentil seedlings exposed to CuO NPs at (**a**,**b**) 100 mg/L (**c**,**d**) 25 mg/L. The *t*-test was used to determine statistical significance; * *p* < 0.05, ** *p* < 0.01 compared to the control.

**Figure 7 nanomaterials-15-01214-f007:**
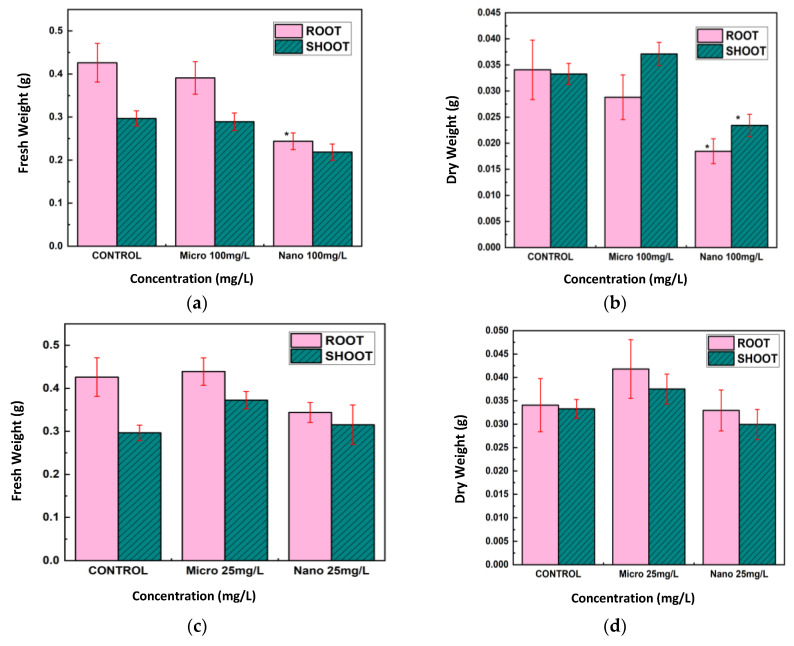
After 7 days, the fresh and dry weight of the roots and shoots of lentil seedlings exposed to Cu metal NPs at (**a**,**b**) 100 mg/L and (**c**,**d**) 25 mg/L were measured. The *t*-test was used to establish statistical significance; * *p* < 0.05 compared to the control.

**Figure 8 nanomaterials-15-01214-f008:**
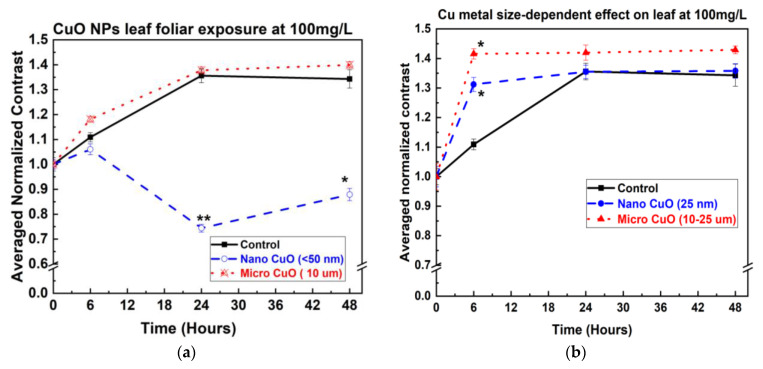
Averaged normalized contrast for leaf exposed to nano and micro particles at 100 mg/L (**a**) CuO, (**b**) Cu metal. *t*-test; * *p* < 0.05, ** *p* < 0.01 vs. control (statistically significant). Legend colors/symbols represent particle sizes: black squares = control, red cross triangles = MPs, blue open circle = NPs.

**Table 1 nanomaterials-15-01214-t001:** Summary of the effects of Cu-based NPs and MPs at 25 and 100 mg/L on bOCT (measured after 20 h) and the conventional physiological and biochemical parameters (measured after 7 days) in lentil seeds. Effects are shown relative to the control.

Particle Type	Concentration (mg/L)	Seed bOCT Contrast (24 h)	Leaf bOCT Contrast (24 h)	Seedling Length (cm) (7 d)	Seedling Biomass (g) (7 d)	CAT Activity (U/g·min) (7 d)
CuO NPs (<50 nm)	25	Decrease	-	Decrease	No change	No change
	100	Decrease	Decrease **	Decrease *	Decrease *	Increase *
CuO MPs (<10 µm)	25	Increase *	-	Increase *	Increase *	Decrease
	100	No change	No change	Increase	Decrease	No change
Cu NPs (25 nm)	25	Decrease	-	Decrease	No change	Increase
	100	Decrease *	Increase	Decrease *	Decrease *	Increase *
Cu MPs (10–25 µm)	25	Increase *	-	Increase	Increase	Decrease *
	100	Increase *	Increase	No change	No change	No change

Asterisk (*) shows a statistically significant variation from the control (*t*-test, * *p* < 0.05, ** *p* < 0.01).

## Data Availability

The data presented in this study are available on request from the corresponding author.

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
