# Peer review of "Rapid Size-Dependent Impact of Cu and CuO Nanoparticles on Lentil Seeds and Leaves Using Biospeckle Optical Coherence Tomography"

_nanomaterials, 2025, doi:10.3390/nano15161214_

Round 1

Reviewer 1 Report

Comments and Suggestions for Authors

This manuscript studies the phytotoxic effects of Cu/CuO NPS and MPs on lentil seeds using bOCT, which is a novel approach for non-invasive, real-time monitoring of early physiological responses. Several concerns should be addressed before the manuscript is accepted for publication.

  1. The authors are encouraged to provide electron microscopy images of Cu nanoparticles/microparticles. What is the size distribution of Cu/CuO NPs? For NPs of 10 nm vs. 50 nm, do they behave differently?
  2. While bOCT offers new insights into early physiological changes, it would be helpful to provide more information on the resolution limits and potential artifacts associated with this imaging technique, particularly in comparison to established methods.
  3. Please discuss the biological mechanisms through which Cu and CuO nanoparticles induce oxidative stress in plants.
  4. In Fig. 8, the authors mentioned the differing effects of Cu NPs versus CuO NPs on leaves. Please discuss the potential reasons for these contrasts.

Author Response

Response to Reviewer Comments

Manuscript ID: nanomaterials 3734645

Title: Rapid size-dependent impact of Cu and CuO nanoparticles on lentil seeds and leaves using biospeckle OCT

Reviewer 1

This manuscript studies the phytotoxic effects of Cu/CuO NPS and MPs on lentil seeds using bOCT, which is a novel approach for non-invasive, real-time monitoring of early physiological responses. Several concerns should be addressed before the manuscript is accepted for publication.

  1. The authors are encouraged to provide electron microscopy images of Cu nanoparticles/microparticles. What is the size distribution of Cu/CuO NPs? For NPs of 10 nm vs. 50 nm, do they behave differently?

Response: Thank you for your valuable comments and suggestions. In our study, we utilized CuO particles (trace metal basis) with primary particle sizes of <50 nm (TEM, ≥99.5%) and <10 µm (≥99.9% trace metals basis), as well as Cu metal particles of 25 nm and 10-25 µm, all sourced as dry powders from Merck Sigma Aldrich, Japan. The detailed properties of each type of nanoparticle (NP), as provided in the Certificate of Analysis from Sigma-Aldrich, together with scanning electron microscopy (SEM) images of the particles, are included in Appendix A.

Regarding the size distribution and their effects, our results indicate that smaller-sized particles (<50 nm for CuO and 25 nm for Cu) consistently exhibited a negative impact on seed germination. In contrast, Cu metal nanoparticles had a significant positive impact on leaf development. Notably, the larger Cu particles (10-25 µm), which are spheroidal in shape, demonstrated a significant positive effect on seed germination compared to both the 25 nm Cu particles and CuO nanoparticles.

These findings suggest that particle size plays a crucial role in determining the biological response, with larger Cu particles being more beneficial for seed germination, while smaller particles may inhibit this process. The detailed data and representative SEM images supporting these observations are provided in Appendix A for your reference.

  1. While bOCT offers new insights into early physiological changes, it would be helpful to provide more information on the resolution limits and potential artifacts associated with this imaging technique, particularly in comparison to established methods.

Response: Thank you for your comment regarding the resolution limits and potential artifacts of bOCT. We would like to clarify that, unlike conventional OCT where speckle is typically regarded as a noise artifact that can obscure structural details, in our bOCT approach, the speckle patterns are intentionally analyzed as a signal source. The temporal and spatial fluctuations of speckle in our system are directly related to internal physiological activity within the biological sample, allowing for early detection of physiological responses to environmental stressors such as nanoparticle exposure. Thus, speckle in bOCT carries functional information rather than being treated solely as noise. However, we acknowledge that random speckle noise and system-related artifacts (such as sample movement, instrumental noise, or optical misalignment) can still be present and may affect the interpretation of results.

To address this, we have implemented appropriate controls and image processing strategies to distinguish physiologically relevant speckle changes from background noise. To further minimize confounding effects, regions of interest were carefully selected between the cotyledon and seed coat, avoiding surface-adjacent or highly reflective areas, as described in the Methods section. The axial and lateral resolution limits of our system (6 µm and 39 µm, respectively) are determined by the optical configuration and are comparable to those of conventional swept-source OCT systems.

While bOCT offers lower spatial resolution than confocal or electron microscopy, it provides unique, non-invasive access to dynamic, functional information in intact biological tissues and enables rapid, repeated, and in situ monitoring of internal activity. In response to your suggestion, we have added the following paragraph to the revised manuscript to discuss potential artifacts. (Pg 4, Ln 165-178)

  1. Please discuss the biological mechanisms through which Cu and CuO nanoparticles induce oxidative stress in plants.

Response: Thank you for this important suggestion. In response, we have added a detailed paragraph to the Discussion section describing the biological mechanisms by which Cu and CuO nanoparticles induce oxidative stress in plants. The added text explains how nanoparticle uptake leads to copper ion release and ROS generation, resulting in oxidative damage to cellular components and inhibition of growth. We also discuss the upregulation of antioxidant enzymes such as CAT as a protective response and how excessive ROS can overwhelm these defenses. Relevant literature has been cited to support this mechanism. (Pg 16, Ln 454-475)

  1. In Fig. 8, the authors mentioned the differing effects of Cu NPs versus CuO NPs on leaves. Please discuss the potential reasons for these contrasts.

Response: Thank you for highlighting the need to discuss the differing effects of Cu and CuO nanoparticles on lentil leaves. In the revised Discussion section, we have added a paragraph addressing this point. (Pg17, Ln 476- 492)

In our study, Cu NPs applied as foliar exposure at 100 mg/L promoted bOCT contrast in lentil leaves, while CuO NPs at the same concentration induced negative effects. The observed differences in our work may be attributed to several factors:

  • Nanoparticle Chemistry and Reactivity: Cu NPs can act as a direct micronutrient source, supplying bioavailable copper that supports essential physiological processes, especially in species or tissues with latent micronutrient demand. In contrast, CuO NPs, due to their higher oxidative potential and tendency to generate reactive oxygen species (ROS) upon foliar contact, may more readily induce oxidative stress and cellular damage in lentil leaves at this concentration.
  • Species-Specific Response: Lentil may possess a higher tolerance or more effective detoxification mechanisms for Cu NPs, while being more sensitive to CuO NPs. The antioxidant response and cellular repair pathways can differ markedly between species and even cultivars.
  • Particle Size and Dissolution: The positive effect of micro-sized Cu and the lack of significant effect from micro-sized CuO in our results further support the importance of particle size and dissolution rate. Nanoscale particles have higher reactivity and bioavailability, which can amplify both beneficial and toxic effects depending on the copper form and plant context.

Given these factors, our findings highlight the importance of plant species, exposure route, and nanoparticle characteristics in determining the outcome of nanoparticle-plant interactions.

Reviewer 2 Report

Comments and Suggestions for Authors

The authors address a topic of interest, such as the interaction of metallic nanoparticles on processes such as germination and leaf area. The information is relevant, and they present their results in a very coherent, clear, and specific manner. However, there are some areas for improvement. 1. The introduction is relevant to the topic and provides relevant information about the group's work. 2. The methodology is consistent; however, the hydrogen peroxide used for seed disinfection can induce effects on the germination process. Furthermore, no treatment without this compound is observed. Therefore, it is unknown or unjustified whether the use of peroxide combined with the interaction of nanoparticles could generate the information presented in the document. It is recommended to include information justifying the use of peroxide, as well as ruling out its possible interaction with the results. 3. In Figure 5, include a reference scale in sections c and f. 4.- Table 1 suggests including the units for each of the measured variables. 5.- The discussion and conclusions are consistent with the results obtained and presented in the manuscript.

Author Response

Response to Reviewer Comments

Manuscript ID: nanomaterials 3734645

Title: Rapid size-dependent impact of Cu and CuO nanoparticles on lentil seeds and leaves using biospeckle OCT

Reviewer 2

The authors address a topic of interest, such as the interaction of metallic nanoparticles on processes such as germination and leaf area. The information is relevant, and they present their results in a very coherent, clear, and specific manner. However, there are some areas for improvement. 1. The introduction is relevant to the topic and provides relevant information about the group's work. 2. The methodology is consistent; however, the hydrogen peroxide used for seed disinfection can induce effects on the germination process. Furthermore, no treatment without this compound is observed. Therefore, it is unknown or unjustified whether the use of peroxide combined with the interaction of nanoparticles could generate the information presented in the document. It is recommended to include information justifying the use of peroxide, as well as ruling out its possible interaction with the results. 3. In Figure 5, include a reference scale in sections c and f. 4.- Table 1 suggests including the units for each of the measured variables. 5.- The discussion and conclusions are consistent with the results obtained and presented in the manuscript.

Response: We thank the reviewer for their positive feedback and constructive suggestions. Please find our responses to each point below:

  1. Introduction:
    We appreciate your acknowledgment of the relevance and clarity of our introduction.

  1. Methodology – Use of Hydrogen Peroxide:

Thank you for your observation regarding the use of hydrogen peroxide (Hâ‚‚Oâ‚‚) in seed disinfection. We used a low concentration (2.5%) for a short duration (10 minutes), followed by three thorough rinses with distilled water to ensure the removal of any residual Hâ‚‚Oâ‚‚. This protocol is widely used and minimizes any direct effect on germination, as Hâ‚‚Oâ‚‚ rapidly decomposes into water and oxygen. Since all treatments, including the control, underwent the same disinfection process, any observed effects can be attributed to the nanoparticle treatments rather than to Hâ‚‚Oâ‚‚ exposure. (Pg 3, Ln 130-132)

  1. Figure 5 – Reference Scale:

We have added a reference scale to sections c and f of Figure 5 as requested.

  1. Table 1 – Units:

Units for all measured variables have been included in Table 1.

  1. Discussion and Conclusions:

We appreciate your positive assessment of our discussion and conclusions and have made minor clarifications as needed.

Thank you again for your valuable feedback, which has helped us improve the quality and clarity of our manuscript.

Reviewer 3 Report

Comments and Suggestions for Authors

I like this MS, L. Tyagi et al. „ Rapid size-dependent impact of Cu and CuO nanoparticles on lentil seeds and leaves using biospeckle OCT” (ID: nanomaterials-3734645).

The idea of using bOCT for answering this research question is excellent, really up to date. It is used appropriately, and the data evaluation is correct. The topic fits well with the experience and the publication history of the authors. MS also fits the scope of the journal and would meet the interests of possible readers.

I have only a couple of suggestions for minor revision, mainly formal notes (more or less in the order of appearance in the MS) as follows.

Make uniform the characters of figures.

Please define ANC (e.g. refer to Eq. 1 ????).

Ls 105-106: Please, provide parameters of sonication (e.g. power, time, mode).

Fig. 5: “Length (cm.) – delete dot.

Figs 6 and 7: “Dry weight (gm)” and “Dry weight (gm.)” – If it is “gram” the sigh is simply “g”. In case of milligram, it should be “mg”. Again…, no dot at the end.

In conclusion, MS is definitely interesting for possible readers of the journal, I definitely suggest publishing after minor revision.

Author Response

Response to Reviewer Comments

Manuscript ID: nanomaterials 3734645

Title: Rapid size-dependent impact of Cu and CuO nanoparticles on lentil seeds and leaves using biospeckle OCT

Reviewer 3

I like this MS, L. Tyagi et al. „ Rapid size-dependent impact of Cu and CuO nanoparticles on lentil seeds and leaves using biospeckle OCT” (ID: nanomaterials-3734645).

The idea of using bOCT for answering this research question is excellent, really up to date. It is used appropriately, and the data evaluation is correct. The topic fits well with the experience and the publication history of the authors. MS also fits the scope of the journal and would meet the interests of possible readers.

I have only a couple of suggestions for minor revision, mainly formal notes (more or less in the order of appearance in the MS) as follows.

Make uniform the characters of figures.

Please define ANC (e.g. refer to Eq. 1 ????).

Ls 105-106: Please, provide parameters of sonication (e.g. power, time, mode).

Fig. 5: “Length (cm.) – delete dot.

Figs 6 and 7: “Dry weight (gm)” and “Dry weight (gm.)” – If it is “gram” the sigh is simply “g”. In case of milligram, it should be “mg”. Again…, no dot at the end.

In conclusion, MS is definitely interesting for possible readers of the journal, I definitely suggest publishing after minor revision.

Response: We sincerely thank the reviewer for their positive and encouraging feedback on our manuscript, as well as for the constructive suggestions to improve the clarity and consistency of our work. We have carefully addressed all the points raised:

  1. Uniform Characters in Figures:

All figure characters have been standardized for consistency throughout the manuscript.

  1. Definition of ANC:

Thank you for your comment. We have clarified the definition of ANC (Averaged Normalized Contrast) in the revised manuscript. Specifically, ANC is calculated in a two-step process:

  1. Contrast Calculation: For each region of interest (ROI), the contrast is calculated using Eq. 1.

  1. Normalization and Averaging: The average local contrast for each treatment and time point is normalized to the pre-imbibition (0 h) value for each seed. The averaged normalized contrast (ANC) for each treatment is then obtained by averaging these normalized contrast values over all seeds in the sample.

We have updated the manuscript to include this stepwise definition and to explicitly refer to Eq. 1 as the basis for the initial contrast calculation. (Pg 7, Ln 267-270)

  1. Sonication Parameters (Lines 105–106):

The sonication parameters (power, time, and mode) have been added to the relevant section for clarity. (Pg 3, Ln 119)

  1. Figure 5 - Length (cm):

The extraneous dot after “cm” in Figure 5 has been removed.

  1. Figures 6 and 7 - Dry Weight Units:

The unit for dry weight has been corrected to “g” (gram) where appropriate, and all unnecessary dots have been removed.

We appreciate the reviewer’s helpful comments, which have improved the quality and presentation of our manuscript. We hope the revised version meets your expectations.

Reviewer 4 Report

Comments and Suggestions for Authors

This work reports the benefit of Cu and CuO NPs for the internal physiological activity of lentil seeds and the observation was made by bOCT. The result indicates that the addition of Cu-based NPs leads to positive effect to lentil seeds and there is also size-specific effect. Though this work is intriguing to relate nanoscience to agricultural science, there is some improvement still needs to be made as below.

  1. Some basic materials characterizations (SEM, TEM, XRD) for Cu and CuO NPs should be presented to better relate the material to the effect.
  2. The scientific evidence showing that the impurities are removed after the sample preparatoin should be presented and discussed.
  3. What would be the effect of untreated raw Cu and CuO NPs on internal physiological activity of lentil seed? Is there even a difference as compared to the cleaned NPs?
  4. Other than the size effect of NPs, morphological effects are also very common. This point should be discussed with relevant previous literature (e.g., Applied Physics Reviews 11 (4), 041317).
  5. Proposed chemical pathways/mechanism of Cu influencing the internal physiological activity of lentil seeds should be presented and discussed.

Author Response

Response to Reviewer Comments

Manuscript ID: nanomaterials 3734645

Title: Rapid size-dependent impact of Cu and CuO nanoparticles on lentil seeds and leaves using biospeckle OCT

Reviewer 4

This work reports the benefit of Cu and CuO NPs for the internal physiological activity of lentil seeds and the observation was made by bOCT. The result indicates that the addition of Cu-based NPs leads to positive effect to lentil seeds and there is also size-specific effect. Though this work is intriguing to relate nanoscience to agricultural science, there is some improvement still needs to be made as below.

  1. Some basic materials characterizations (SEM, TEM, XRD) for Cu and CuO NPs should be presented to better relate the material to the effect.

Response: Thank you for this important suggestion. In response, we have included detailed materials characterization data for all Cu and CuO particles used in this study. The primary particle sizes (as specified by the manufacturer), as well as scanning electron microscopy (SEM) images, are now provided in Appendix A. These data are based on the Certificate of Analysis from Sigma-Aldrich and our own SEM analysis. This addition allows for a clearer correlation between the physical properties of the materials and their observed effects on lentil plants.

  1. The scientific evidence showing that the impurities are removed after the sample preparation should be presented and discussed.

Response: Thank you for your comment. The Cu and CuO nanoparticles used in our study were purchased from Sigma-Aldrich, a reputable supplier providing high-purity nanomaterials (typically more than 95% purity of NPs as per Certificate of Analysis). To further minimize potential impurities and ensure a homogeneous suspension, we sonicated the nanoparticles in distilled water containing Tween 80, a non-ionic surfactant widely used to prevent aggregation and aid dispersion. While we did not perform additional purification or direct impurity analysis post-sonication, the use of commercially available, analytically certified nanoparticles is a standard practice in nanotoxicology studies. Sigma-Aldrich’s quality control includes XRD and chemical purity analysis, which confirms the absence of significant impurities in the supplied material. We acknowledge this as a limitation and added a statement in the manuscript discussing the reliance on supplier-provided purity data and the absence of further post-preparation impurity analysis. (Pg 3, Ln 121-125)

  1. What would be the effect of untreated raw Cu and CuO NPs on internal physiological activity of lentil seed? Is there even a difference as compared to the cleaned NPs?

Response: Thank you for your question. In our study, we used Cu and CuO nanoparticles as received from Sigma-Aldrich, dispersing them in water with Tween 80 and sonication, but without any additional chemical cleaning or purification steps. The term “cleaned” was not used in our manuscript, and we did not perform any such treatment on the nanoparticles. Commercially available, analytical-grade nanoparticles from Sigma-Aldrich have low and well-characterized impurity levels. Therefore, the physiological effects observed in our experiments can be attributed to the nanoparticles themselves rather than to potential contaminants. While the presence of impurities in non-commercial or laboratory-synthesized nanoparticles can influence plant responses, this is unlikely in our case. We have clarified this point in the revised manuscript. We also acknowledge that future studies could compare “as received” and further purified nanoparticles to directly address this question.

  1. Other than the size effect of NPs, morphological effects are also very common. This point should be discussed with relevant previous literature (e.g., Applied Physics Reviews 11 (4), 041317).

Response: Thank you for your valuable suggestion. In response, we have expanded the Discussion section to address the impact of nanoparticle morphology (shape) on plant responses, in addition to size effects. We discussed how different shapes, such as spheroidal versus irregular or rectangular particles, can influence nanoparticle uptake, distribution, and physiological effects in plants. This section references recent literature to support our discussion and to relate these findings to our results with Cu and CuO particles in lentil. (Pg 17, Ln 503-530)

  1. Proposed chemical pathways/mechanisms of Cu influencing the internal physiological activity of lentil seeds should be presented and discussed.

Response: Thank you for your suggestion. In response, we have added a detailed discussion of the proposed chemical pathways and mechanisms by which Cu and CuO nanoparticles influence the internal physiological activity of lentil seeds. This new section is included in the Discussion (Pg 16, Ln 454- 475). The added text explains how these nanoparticles penetrate seed tissues, release copper ions, and catalyze redox reactions that generate reactive oxygen species (ROS), leading to oxidative stress, activation of antioxidant defenses such as catalase (CAT), and, when ROS levels exceed cellular defenses, resulting in growth inhibition and cellular damage. Relevant literature has been cited to support these mechanisms.

Round 2

Reviewer 1 Report

Comments and Suggestions for Authors

The authors have addressed the questions raised by the reviewer, and I am satisfied with the modifications they have made to the manuscript.

The positioning of figure labels, such as a, b, c, etc., can be adjusted to meet the journal's requirements.

Reviewer 2 Report

Comments and Suggestions for Authors

The authors have addressed the suggestions made to the document, which allows us to appreciate a more enriched manuscript in line with the quality required by the journal.    

Reviewer 4 Report

Comments and Suggestions for Authors

The revised manuscript can be accepted.